# Crown-Level Structure and Fuel Load Characterization from Airborne and Terrestrial Laser Scanning in a Longleaf Pine (*Pinus palustris* Mill.) Forest Ecosystem

Kleydson Diego Rocha [1,*], Carlos Alberto Silva [1], Diogo N. Cosenza [1], Midhun Mohan [2],
Carine Klauberg [1], Monique Bohora Schlickmann [1], Jinyi Xia [1], Rodrigo V. Leite [1],
Danilo Roberti Alves de Almeida [3], Jeff W. Atkins [4], Adrian Cardil [5], Eric Rowell [6], Russ Parsons [7],
Nuria Sánchez-López [8], Susan J. Prichard [9] and Andrew T. Hudak [10]

1. Forest Biometrics, Remote Sensing and Artificial Intelligence Laboratory (Silva Lab), School of Forest, Fisheries, and Geomatics Sciences, University of Florida, Gainesville, FL 32611, USA
2. Department of Geography, University of California—Berkeley, Berkeley, CA 94709, USA
3. Department of Forest Sciences, "Luiz de Queiroz" College of Agriculture (USP/ESALQ), University of São Paulo, Piracicaba 13418-900, SP, Brazil
4. Southern Research Station, USDA Forest Service, Savannah River Site, New Ellenton, SC 29809, USA
5. Tecnosylva, Parque Tecnológico de León, 24009 León, Spain
6. Division of Atmospheric Sciences, Desert Research Institute, 2215 Raggio Parkway, Reno, NV 89512, USA
7. Fire Sciences Laboratory, Rocky Mountain Research Station, USDA Forest Service, 5775 W. Highway 10, Missoula, MT 59801, USA
8. Department of Forest, Rangeland and Fire Sciences, College of Natural Resources, University of Idaho, 875 Perimeter Drive, Moscow, ID 83844, USA
9. School of Environmental and Forest Sciences, University of Washington, Seattle, WA 98195, USA
10. Forestry Sciences Laboratory, Rocky Mountain Research Station, USDA Forest Service, 1221 South Main Street, Moscow, ID 83843, USA
* Correspondence: kdarocha@ufl.edu; Tel.: +1-(352)-213-5937

**Abstract:** Airborne Laser Scanners (ALS) and Terrestrial Laser Scanners (TLS) are two lidar systems frequently used for remote sensing forested ecosystems. The aim of this study was to compare crown metrics derived from TLS, ALS, and a combination of both for describing the crown structure and fuel attributes of longleaf pine (*Pinus palustris* Mill.) dominated forest located at Eglin Air Force Base (AFB), Florida, USA. The study landscape was characterized by an ALS and TLS data collection along with field measurements within three large (1963 m$^2$ each) plots in total, each one representing a distinct stand condition at Eglin AFB. Tree-level measurements included bole diameter at breast height (DBH), total height (HT), crown base height (CBH), and crown width (CW). In addition, the crown structure and fuel metrics foliage biomass (FB), stem branches biomass (SB), crown biomass (CB), and crown bulk density (CBD) were calculated using allometric equations. Canopy Height Models (CHM) were created from ALS and TLS point clouds separately and by combining them (ALS + TLS). Individual trees were extracted, and crown-level metrics were computed from the three lidar-derived datasets and used to train random forest (RF) models. The results of the individual tree detection showed successful estimation of tree count from all lidar-derived datasets, with marginal errors ranging from −4 to 3%. For all three lidar-derived datasets, the RF models accurately predicted all tree-level attributes. Overall, we found strong positive correlations between model predictions and observed values ($R^2$ between 0.80 and 0.98), low to moderate errors (RMSE% between 4.56 and 50.99%), and low biases (between 0.03% and −2.86%). The highest $R^2$ using ALS data was achieved predicting CBH ($R^2$ = 0.98), while for TLS and ALS + TLS, the highest $R^2$ was observed predicting HT, CW, and CBD ($R^2$ = 0.94) and HT ($R^2$ = 0.98), respectively. Relative RMSE was lowest for HT using three lidar datasets (ALS = 4.83%, TLS = 7.22%, and ALS + TLS = 4.56%). All models and datasets had similar accuracies in terms of bias (<2.0%), except for CB in ALS (−2.53%) and ALS + TLS (−2.86%), and SB in ALS + TLS data (−2.22%). These results demonstrate the usefulness of all three lidar-related methodologies and lidar modeling overall, along with lidar applicability in the estimation of crown structure and fuel attributes of longleaf pine forest ecosystems. Given that TLS measurements are less

practical and more expensive, our comparison suggests that ALS measurements are still reasonable for many applications, and its usefulness is justified. This novel tree-level analysis and its respective results contribute to lidar-based planning of forest structure and fuel management.

**Keywords:** lidar; machine learning; crown structure; southern forest; fusion

## 1. Introduction

Forests cover approximately one-third of the earth's total land area and enrich human lives on a day-to-day basis by providing timber, food, fuel, fodder, and bioproducts [1,2]. Apart from their economic significance, forested landscapes also offer numerous ecosystem services such as carbon sequestration, biodiversity conservation, nutrient cycling, and water and air purification, which are crucial for human well-being and global sustainability [3–5]. Therefore, precise and timely estimation of forest growth and structure is considered a critical element in forest management and environmental change assessment. Additionally, information on single trees is required to understand the phenomena and processes occurring in forests [6]. In the case of pine-dominated forests, these tasks are primarily accomplished by employing growth and yield models, which make use of tree attribute information obtained through various direct and indirect data collection strategies [7,8].

Longleaf pine (*Pinus palustris* Mill.), native to the southeastern US, was one of the most extensive ecosystems—with ecological and economic importance—in North America [9]. Compared to other southeastern pines, they are more resilient to the negative impacts of global climate change and can withstand severe windstorms, resist pests, tolerate drought, and help reduce air pollution [10,11]. Additionally, they support the existence of numerous animals and birds that rely on longleaf pines for their habitat [12,13]. However, due to their high-quality wood, these forests were heavily harvested for lumber production and construction purposes throughout the 19th century. This, in addition to land-use changes and fire suppression regimes, has resulted in the depletion of these landscapes, from over 92 million acres to a mere 3 million acres in a few decades. That being the case, sustainable management and conservation of existing longleaf forests take high priority, and for this, accurate measurement of tree counts and other forest attributes that are used at the tree, plot, and stand levels are vital.

While many forest attributes are made using traditional field sampling techniques, individual tree field measurements over large areas are generally non-viable due to limitations set forth by budget, time, and labor constraints. As a consequence, remote sensing technology, in particular Light Detection and Ranging (lidar), has become the most sought-after remote sensing technology for plot and stand-level forest inventory due to its ability to provide highly accurate and spatially detailed information about forest attributes across vast forested landscapes [14,15]. With the boom of increased dataset availability, advanced data processing capabilities, and algorithmic developments, applications of lidar technologies in forestry are expected to expand multiple folds in the coming years. Among the lidar systems, Airborne Laser Scanners (ALS) and Terrestrial Laser Scanners (TLS) are implemented extensively in the forest management discipline due to their ability to characterize forest vertical structure at the stand or plot level using the acquired high-density 3D point clouds [8,16,17].

ALS can be used to acquire the vertical and horizontal forest structure in detail with laser pulses [18], and due to cost-efficiency and scale of operation, ALS has received by far the most attention for large area retrieval of forest structural parameters [19]. In particular, ALS has been applied for individual tree detection (ITD) over large areas, as well as for tree crown delineation, forest uniformity calculation, and canopy gap analysis [20–23]. Although airborne systems can cover large areas efficiently and relatively cost-effectively [24,25], difficulties persist in the observation of near-ground vegetation and the lower canopy characteristics [26]. In addition, the estimation of forest structural param-

eters using ALS faces a few persistent issues caused by factors such as species heterogeneity, tree density, and crown overlapping that are not simply resolved by increased lidar return density [8,13,16,27]. For instance, Silva et al. [8] realized the difficulty of modeling tree basal area due to the lack of linearity in the relationship between height and diameter of longleaf pine after this species reaches a diameter of ∼25 cm because longleaf pine height growth asymptotes at ∼25 m while the diameter continues to increase.

Because of these ALS limitations, TLS data can be considered a satisfactory alternative for more completely characterizing three-dimensional (3D) forest structures, especially for the canopy understory given its horizontal and upward view perspectives [17], which complements the downward view perspective of ALS [26]. TLS is capable of acquiring levels of detail far beyond what ALS is capable of [28]. TLS offers an approach for calibrating and validating ALS and thereby helps improve the accuracy of the estimated forest structural variables [16,29–31]. However, TLS data collection methods suffer from their inability to capture information outside their direct line of sight with respect to a fixed location. This results in a high percentage of laser pulses being intercepted by the understory, stems, and lower canopy, reducing the number of pulses reaching the upper canopy and areas that are directly behind the trees [17].

Given that there exist several advantages and limitations for TLS and ALS for improving forest management operations, it is equally important to understand how each data collection strategy affects the accuracy of extracted tree features as well as how certain metrics derived from these tree parameters influence the performance of operational forest modeling systems. In the case of fire behavior and effects models, in general, several tree-level or stand-level parameters—such as tree count and maximum tree height—are initially estimated from remotely sensed data [32,33]. Consequently, these parameters are incorporated into statistical or machine learning models to choose the best metrics for deriving crown-level metrics (for example, crown area and crown height). Finally, these metrics are utilized to develop fire models that can be used for evaluating dynamic relationships existing within the macroscale forest ecosystems, designing prescribed burn plans, and quantifying canopy fuels, which in turn constrains fire behavior [34–36]. The effectiveness and applicability of fire behavior and effects models rely predominantly on the accuracy of forest-related estimates that are provided as inputs. Therefore, it is crucial to develop synergies between complementary datasets—in this case, TLS and ALS—to better estimate tree-level parameters with high accuracy. This way, we can expand the number of obtainable crown structure and fuel estimates from TLS or ALS or their combination. Moreover, analyzing how the accuracy of crown structure and fuel estimation impacts fire modeling outputs (from widely used tools such as QUIC-Fire or WFDS) would allow us to develop ways to streamline the parameters/workflow, dissect complex interactions between individual parameters affecting wildfire behavior, optimize the fire model outputs, and interpret results more constructively [34,37]. The study assesses the utility of ALS and TLS systems and their combination (ALS + TLS) in estimating tree crown structure and fuel attributes in a longleaf pine forest ecosystem. In particular, we used random forest (RF) modeling to compare the capability of ALS, TLS, and ALS + TLS in (i) detecting individual trees from 3D point clouds; (ii) deriving and comparing crown-level metrics from ALS, TLS, and their fusion; and (iii) predicting tree-level structure and fuel attributes, namely stem diameter at breast height (DBH), total tree height (HT), crown base height (CBH), crown width (CW), foliage biomass (FB), stem branches biomass (SB), crown biomass (CB), and crown-bulk density (CBD).

## 2. Materials and Methods

### 2.1. Study Area

The study area is located at Eglin Air Force Base north of Niceville, Florida, USA (Figure 1). The climate is characterized by warm, humid summers, and generally mild winters. The forest type is dominated by longleaf pine (*Pinus palustris* Mill.) and turkey

oaks [38,39]; open overstory canopy cover (up to 50%) and open understories (sparse understory shrub cover) are maintained by frequent prescribed fires.

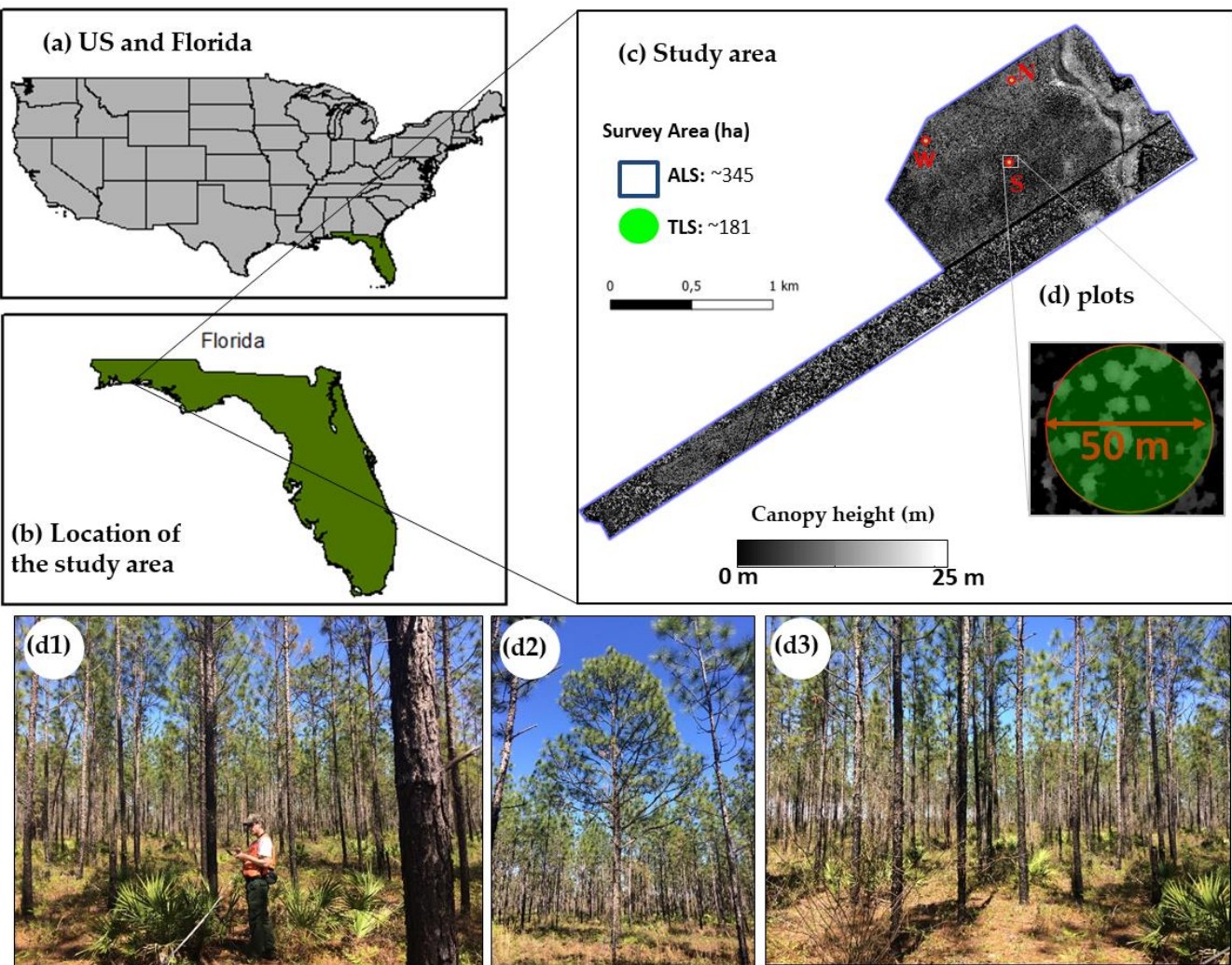

**Figure 1.** Study area and field plot location at Eglin Air Force Base: (**a**) United States, (**b**) state of Florida, and (**c**) ALS and TLS extent. (**d**) Field plots and (**d1–d3**) photos at the bottom were collected during the 2017 field measurement survey.

*2.2. Field Data Collection*

In the field, tree measurements were collected within three circular plots of 25 m radius (0.196 ha) in 2017. Tree stems were geolocated using a differential Global Positioning System (DGPS) (Trimble Geo7X [40]). We used steel diameter tapes to measure the diameter at breast height (DBH, 1.37 m above ground) of all individuals ≥ 5 cm DBH. Laser rangefinders (LaserTech Impulse 200 [41]) were used to measure total tree height (HT, m) and crown base height (CBH, m), and a cloth tape was used to measure crown width (CW, m) along orthogonal (major and minor; $CW_{MJ}$ and $CW_{MN}$) crown axes. All trees tallied (n = 353) were geolocated using a GNSS (Trimble Geo7X); further details can be found in the US Forest Service Research Data Archive [42]. Multi-stem trees (n = 5), dead trees (n = 21), or trees < 10 cm DBH (n = 123) were dropped from consideration, leaving n = 205 trees for subsequent analysis; of these, 146 of 149 conifers were longleaf pine, and 50 of 56 hardwoods were turkey oak, leaving insufficient diversity to consider beyond conifer vs. hardwood trees. Foliage biomass (FB; kg), stem branches biomass (SB; kg), crown biomass (CB; kg), and crown bulk density (CBD; kg m$^{-3}$) were calculated for each

longleaf pine and oak tree based on allometric Equations (1)–(8) below. Table 1 summarizes all the field-based individual tree measurements used in this study.

**Table 1.** Descriptive statistics of field measurements of 205 trees included in the study: height (Ht); diameter at breast height (DBH); canopy base height (CBH); crown width (CW; major and minor); foliage biomass (FB); stem and branches biomass (SB); crown biomass (CB); and crown bulk density (CBD).

| Attributes | Units | Min. | Mean | Max. | sd |
|---|---|---|---|---|---|
| HT | m | 3.90 | 10.65 | 16.90 | 2.96 |
| DBH | cm | 10.0 | 16.40 | 39.50 | 5.05 |
| CBH | m | 0.30 | 4.51 | 8.50 | 2.32 |
| CW | m | 1.30 | 3.26 | 9.45 | 1.36 |
| FB | kg | 1.72 | 7.25 | 41.02 | 5.54 |
| SB | kg | 22.14 | 102.64 | 703.21 | 85.47 |
| CB | kg | 4.77 | 23.09 | 226.92 | 26.54 |
| CBD | $kg\,m^{-3}$ | 0.03 | 0.25 | 1.11 | 0.25 |

Longleaf pine (*Pinus palustris*) [7]:

$$FB\ (kg) = 0.0697 \times DBH^{2.1631} \times HT^{-0.5569} \tag{1}$$

$$SB\ (kg) = 0.0070 \times DBH^{3.6735} \times HT^{-1.1735} + 0.0273 \times DBH^{1.9745} \times HT^{0.9163} \tag{2}$$

$$CB\ (kg) = FB + 0.0070 \times DBH^{3.6735} \times HT^{-1.1735} \tag{3}$$

Oaks (*Quercus* spp.) [43,44]:

$$FB\ (kg) = 0.0211 \times (\pi \times DBH^2)/4 \tag{4}$$

$$SB\ (kg) = 0.5614 \times (\pi \times DBH^2)/4 \tag{5}$$

$$CB\ (kg) = 1.68513 \times 10^{-5} \times DBH^{2.4767} \times 10^2 \tag{6}$$

where DBH is the diameter at breast height (DBH, 1.37 m above ground) and HT is the total crown height (m). Crown bulk density (CBD; $kg\,m^{-3}$) was calculated for each tree based on the following equation:

$$CBD\ (kg\,m^{-3}) = CB\ (kg)/CV_{ellp}\ (m^3) \tag{7}$$

where CB is the crown biomass (kg) and $CV_{ellp}$ is the crown volume ($m^3$) based on a 3D ellipsoid and calculated as a function of crown width ($CW_{MJ}$ and $CW_{MN}$) and crown length (CL):

$$CV_{ellp}\ (m^3) = [4/3 \times \pi \times (CW_{MJ}/2) \times (CW_{MN}/2) \times CL/2] \tag{8}$$

*2.3. ALS Survey*

The ALS data were acquired in November 2012 by Kucera International using a Leica ALS60 sensor operating in Multiple Pulses in Air (MPiA) modes, with a 20° field of view and 50% sidelap, flying at the height of 1200 m. Additional information on the ALS data collection can be found in the US Forest Service Research Data Archive [45].

*2.4. TLS Survey*

The TLS data were acquired in October 2012 using an Optech ILRIS $3_6$D-HD Scanner mounted on a boom lift pointed downward at an angle of 23° at the height of 20 m above the ground. Additionally, for each plot, the TLS sensor was placed at six different positions at a 20 m distance from the border of the plot and was operated remotely from a tablet computer via a wireless connection. Each of the six scans was configured to have a sampling

resolution of 8 mm in the 20 m range, 2 cm in the 90 m range, and 5.6 cm in the 300 m range. Additional information on TLS data collection can be found in the US Forest Service Research Data Archive [46].

### 2.5. ALS and TLS Data Pre-Processing

The summary of the parameters for both ALS and TLS surveys is displayed in Table 2. The data pre-processing was carried out using LAStools [47] and FUSION/LDV LiDAR [48]. First, ALS ground returns were classified using a progressive triangulated irregular network densification algorithm [49] implemented in lasground [47] (settings: step is 20 m, bulge is 0.5 m, spike is 1 m, and switches are wilderness and ultra_fine) to create a 1-m resolution digital terrain model (DTM). Second, both ALS and TLS returns were converted to above-ground heights by subtracting the elevation values of the ALS-derived DTM. The ALS and TLS normalized point clouds were combined using CloudCompare®(Align and Rotate tools) [50] to create a fused dataset (ALS + TLS).

**Table 2.** ALS and TLS collection parameters.

| Attributes | ALS | TLS |
|---|---|---|
| Point Density | 6.8 points m$^{-2}$ | 68.14 points m$^{-2}$ |
| Pulse rate | 178.6 kHz | 10.0 kHz |
| Scan Altitude | 1200 AGL | 16–27 m AGL |

### 2.6. Individual Tree Detection and Crown-Level Metrics

Individual tree detection (ITD), extraction, and crown metrics computation using ALS, TLS and ALS + TLS datasets was performed in three steps using the lidR [51] and lidRplugins [52] packages in the R environment (R version 4.2.1) [53]. First, individual tree detection was performed using an adapted approach by Eysn et al. [54] and implemented in the lidRplugins package. The method is based on iterative canopy height model generation (CHM; 0.5 m spatial resolution), followed by a sequential detection of potential tree tops step using a local maximum filtering approach within a moving window over various CHMs. Tree tops are then sorted by height and considered a detected tree if there is no detected tree within a given 2D distance (2 m) as well as a given 3D distance (1 m) [54]. Relative difference (RD was used for accessing ITD performance (Equation (9)). Second, the individual tree crowns were delineated based on the Voronoi diagram algorithm [55] developed by Silva et al. [8] and implemented in the lidR package [51]. The segmented crown areas were used to obtain the crown radius (CRAD; m) and canopy projected area (CPA; m$^2$), defined as the area of the circle with a radius equal to CRAD. Lastly, all height returns within crown areas were extracted, and individual crown metrics (Table 3) were computed for each tree detected. The mixtools package [56] was used to compute the crown base height (CBH, m) by subtracting three times the component standard deviation from the component mean of the topmost distribution. Crown volume (CV; m$^3$) and crown surface area (CSA; m$^2$) were calculated as the volume and surface area of the 3D convex hulls derived from all lidar returns within crown segments, respectively. The two-sided Kolmogorov–Smirnov (KS) test was used to test if the distributions of the same crown-level metric from ALS, TLS, and ALS + TLS differed significantly, with significance level of 0.05.

The relative difference (RD) between detected and observed trees was calculated as follows:

$$\text{RD (\%)} = \frac{n_l - n_f}{n_l} \, 100 \tag{9}$$

where $n_l$ is the number of trees detected with lidar and $n_f$ is the number of trees inventoried in the field for each plot.

**Table 3.** ALS-derived crown metrics (Klauberg et al. [57]).

| | Definition | Unit | Abbreviation |
|---|---|---|---|
| Crown Height based metrics | Maximum crown height | m | HMAX |
| | Mean crown height | m | HMEAN |
| | Height standard deviation | m | HSD |
| | Variance of heights | m$^2$ | HVAR |
| | Kurtosis of heights | - | HKUR |
| | Skewness of heights | - | HSKEW |
| | X$^{th}$ percentiles of heights | m | H5TH, H15TH, H20TH, ..., H90TH, H95TH, H99TH |
| Crown morphology based metrics | Crown base height | m | CBH |
| | Crown length (HMAX—CBH) | m | CL |
| | CBH-based crown ratio (100 × CL/CBH) | % | CRAT |
| | Crown radius ($\sqrt{Segment\ Crown\ area/\pi}$) | m | CRAD |
| | Simplified crown projected area ($\pi \times$ CRAD$^2$) | m$^2$ | CPA |
| | Crown convex hull volume | m$^3$ | CV |
| | Crown convex hull surface area | m$^2$ | CSA |
| | Crown density (i.e., the ratio between the number of returns above CBH and the total number of returns) | % | CDEN |
| | HMAX-based crown ratio (100 × CL/HMAX) | % | CRT |
| | Crown form index (100 × CL/(2 × CRAD)) | % | CFI |
| | Crown thickness index (100 × (2 × CRAD)/CL) | % | CTI |
| | Crown spread ratio (100 × (2 × CRAD)/CBH) | % | CSR |

*2.7. Crown-Level Structural and Fuel Load Attributes Modeling*

2.7.1. Linking Field and Lidar Detected Trees

　　Field-inventoried trees and individual trees detected from lidar were linked based on the minimum Euclidean distance ($D_{fl}$) calculated from four multidimensional spaces (Equation (10)). The method works in two general steps; first, a lidar tree is randomly selected, and the $D_{fl}$ is computed for all the trees listed in the field database. The tree in the field database that showed the lower $D_{fl}$ is selected for matching. If two or more field trees had the same $D_{fl}$ values, we randomly selected one for matching, and the remaining trees were returned for the next iteration. After the matching, both lidar and field trees are removed from the database, and the process is repeated until all the lidar and field trees are matched.

$$D_{fl} = \sqrt{\left(X_f - X_l\right)^2 + \left(Y_f - Y_l\right)^2 + \left(\text{HMAX}_f - \text{HMAX}_l\right)^2 + \left(\text{CPA}_f - \text{CPA}_l\right)^2} \qquad (10)$$

where $D_{fl}$ is the distance between the *f*-th tree in the field and the *l*-th tree detected in the lidar data; $X_f$ and $Y_f$, and $X_l$ and $Y_l$, are the xy coordinates (UTM) of the trees in the field and lidar data, respectively; $\text{HMAX}_f$ and $\text{HMAX}_l$ are the field (HT) and lidar-based crown height (HMAX) in m, respectively; and $\text{CPA}_f$ and $\text{CPA}_l$ are the field and lidar-based crown projected area (m$^2$), respectively.

2.7.2. Variable Selection and Random Forest Modeling

　　Pearson's correlation (r) and model improvement ratio (MIR) (e.g., [58–60] were applied to identify the most important ALS, TLS, and ALS + TLS crown-level metrics for predicting DBH, HT, CW, CBH, SB, FB, CB, and CBD according to Silva et al. [61] and Klauberg et al. [57]. To create parsimonious models, we reserved only the crown-level metrics that exhibited r < 0.9 and MIRs ≥ 0.25. We used the two-sided Kolmogorov–Smirnov (KS) statistic [62] in R to compare the distribution of the lidar-derived crown metrics within datasets (ALS, TLS, and ALS + TLS) selected by MIR at a significance level of 5%. After variable selection, the crown-level structure and fuel load attributes of interest were predicted at the tree level using also the RF package [63] embedded in a bootstrap

approach with 500 iterations; ntree was set to 1000, and mtry was set to equal the number of best lidar metrics selected by MIR.

In each bootstrap iteration, we drew 205 times with replacements from the 205 available samples within the three field plots. In this procedure, on average, 45% of the total samples (~70 samples) are not drawn. These samples were subsequently used as holdout samples for independent validation (e.g., [24]). In each bootstrap iteration, the calibrated RF model was applied to predict the crown-level structure and fuel attributes for the holdout samples (out of bag). After 500 iterations, the average of the predictions for each tree observation from the holdout samples is used as the final prediction. $R^2$, absolute and relative RMSE, and bias were computed based on the linear relationship between observed and predicted crown structure and fuel attributes for assessing model accuracy (Equations (11)–(13)).

$$R^2 = \frac{\left[\sum_{i=1}^{n}(y_i - \overline{y})(\hat{y}_i - \overline{\hat{y}})\right]^2}{\sum_{i=1}^{n}(y_i - \overline{y})^2 \sum_{i=1}^{n}(\hat{y}_i - \overline{\hat{y}})^2} \tag{11}$$

$$RMSE = \sqrt{\sum_{i=1}^{n}(\hat{y}_i - y_i)^2/n} \tag{12}$$

$$Bias = \frac{1}{n}\sum_{i=1}^{n}(\hat{y}_i - y_i) \tag{13}$$

where $n$ is the number of trees; $y_i$ and $\hat{y}_i$ are the observed and predicted values for tree $i$; $\overline{y}$ and $\overline{\hat{y}}$ are the observed and predicted mean values for tree $i$; and relative RMSE (%) and biases (%) were calculated by dividing the absolute values (Equations (11) and (13)) by the mean of the observed values. Figure 2 displays an overview of the methodology used in this project.

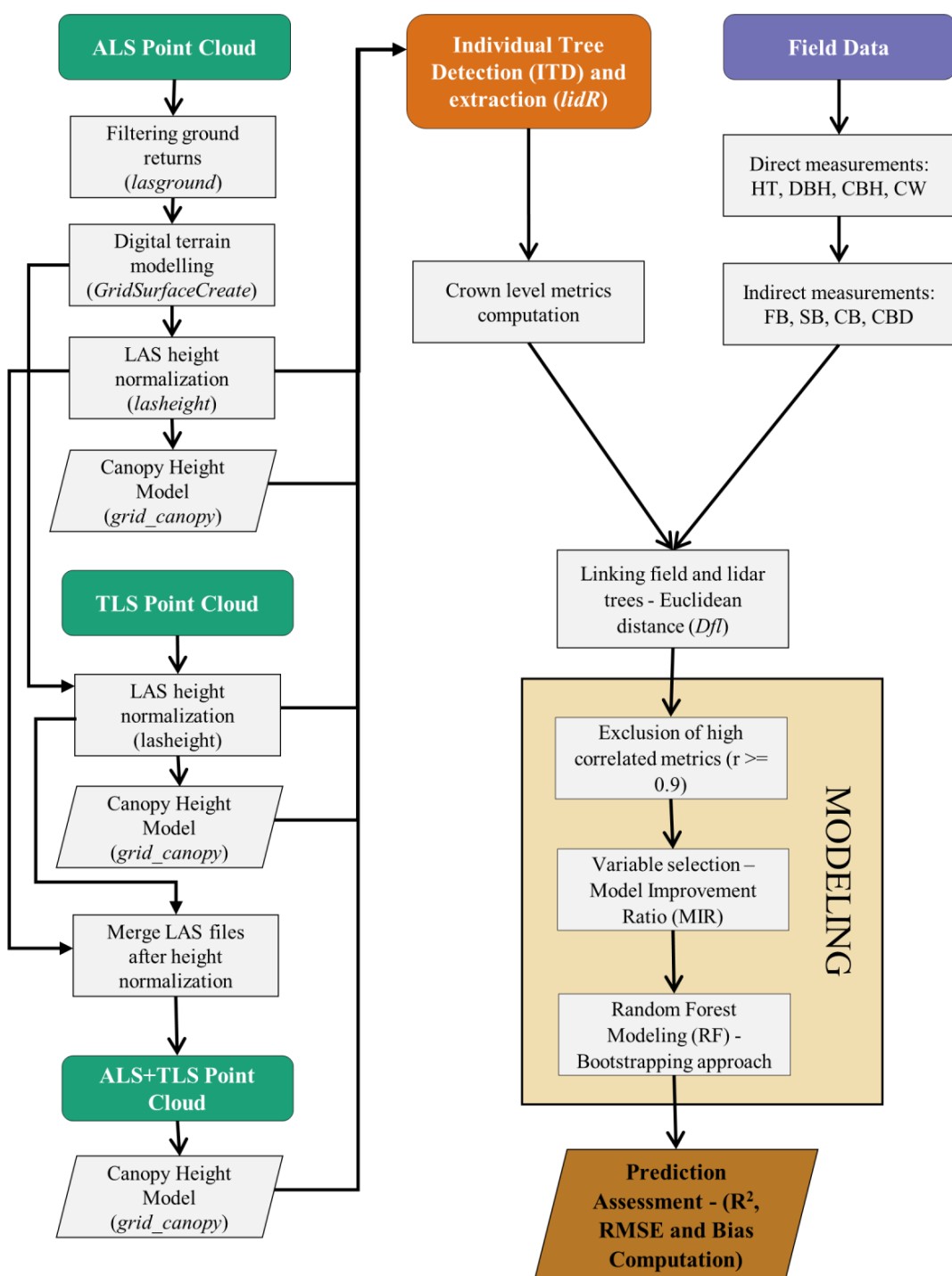

**Figure 2.** Overall workflow for data processing, individual tree detection and extraction, modeling, and prediction assessment.

## 3. Results

### 3.1. Individual Tree Detection and Crown-Level Metrics

The number of trees detected by the ITD algorithm was similar to the number of trees $\geq 10$ cm DBH tallied in the field (Table 4, Figure 3). The relative difference between the number of observed and detected trees relative to the observed trees (i.e., detection error) was between $-4\%$ and $3\%$, which can be considered marginal. There is no clear error pattern across plots or datasets, which suggests that there is no systematic source of errors in the ITD algorithm. Fused data (ALS + TLS) provided the lowest error range

(from 0% to 2%), followed by the ALS data (from −2% to 2%) and TLS (from −4% to 2%). While the overall error range of the ALS data seems slightly smaller when compared to the TLS dataset, only one of the plots (N) was worse in the TLS dataset (possibly as a result of occlusion because of the higher stand density). In the other two plots, the TLS dataset either presented no error (W) or had an error similar to the one registered by the ALS dataset (S). Although these ranges are comparable in practice, this pattern suggests that the ITD algorithm effectiveness benefits slightly from ALS + TLS. Figure 4 shows the distribution of crown-level metrics computed from ALS, TLS, and ALS + TLS datasets. ALS-derived crown metrics in most of the cases differed significantly ($p < 0.05$) from TLS and ALS + TLS-derived crown metrics based on the KS test. However, TLS and ALS + TLS crown metrics were similar in all cases based on the KS test (Figure 4).

**Table 4.** Result of the individual tree detection (ITD) using ALS, TSL, and the fused (ALS + TLS) lidar datasets; n = number of detected trees; RD = relative difference between detected and observed trees relative to the number of observed trees (detection error).

| Plot | Observed | ALS | | TLS | | ALS + TLS | |
|---|---|---|---|---|---|---|---|
| | | n | RD | n | RD | n | RD |
| N | 121 | 124 | 2% | 116 | −4% | 123 | 2% |
| W | 33 | 34 | 3% | 33 | 0% | 33 | 0% |
| S | 51 | 50 | −2% | 52 | 2% | 51 | 0% |

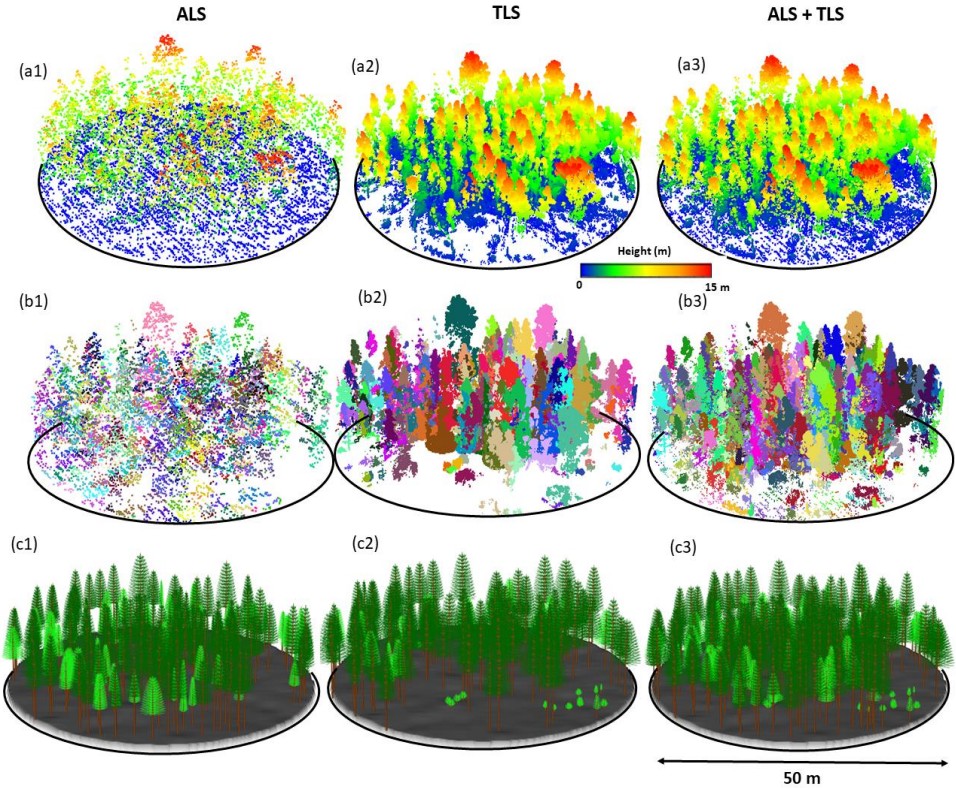

**Figure 3.** Illustration of the individual tree detection (ITD) and extraction from ALS, TLS, and ALS + TLS data in one of the plots: (**a1–a3**) raw point cloud of ALS, TLS, and ALS + TLS data colored by point height above ground; (**b1–b3**) point cloud trees of ALS, TLS, and ALS + TLS segmented by the ITD algorithm; (**c1–c3**) 3D representation of the trees in the plots retrieved from ALS, TLS, ans ALS + TLS data [64].

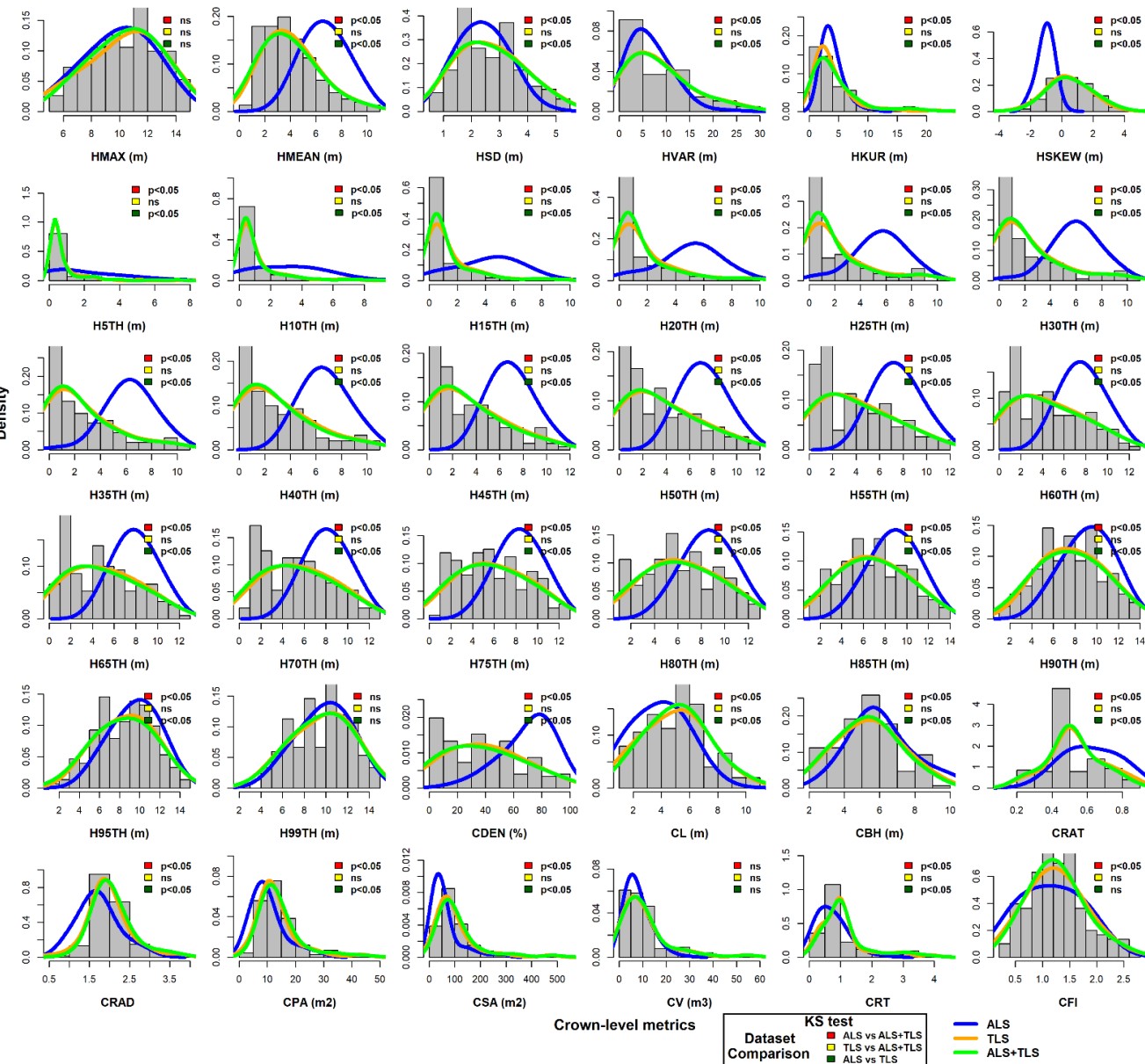

**Figure 4.** Density plot of the ALS (blue line), TLS (orange line), and ALS + TLS (green line) derived crown metrics. The red, yellow, and green squares represent the Kolmogorov–Smirnov test when comparing ALS vs. ALS + TLS, TLS vs. ALS + TLS, and ALS vs. TLS crown metrics distribution, respectively. The gray histogram is generated using the ALS + TLS crown metrics data. Ns refers to not significantly ($p > 0.05$) in the two-sided Kolmogorov–Smirnov (KS) test.

### 3.2. Variable Selection

The first stage of the variable selection approach based on r values considerably reduced the number of candidate metrics for the modeling, from 36 to 15 metrics in the ALS data, to 16 metrics in the TLS data, and to 15 metrics in the ALS + TLS data (Figures 5 and 6). Both crown height and crown morphology-based metrics were selected in the first filtering, although the number of crown morphology-based metrics was greater.

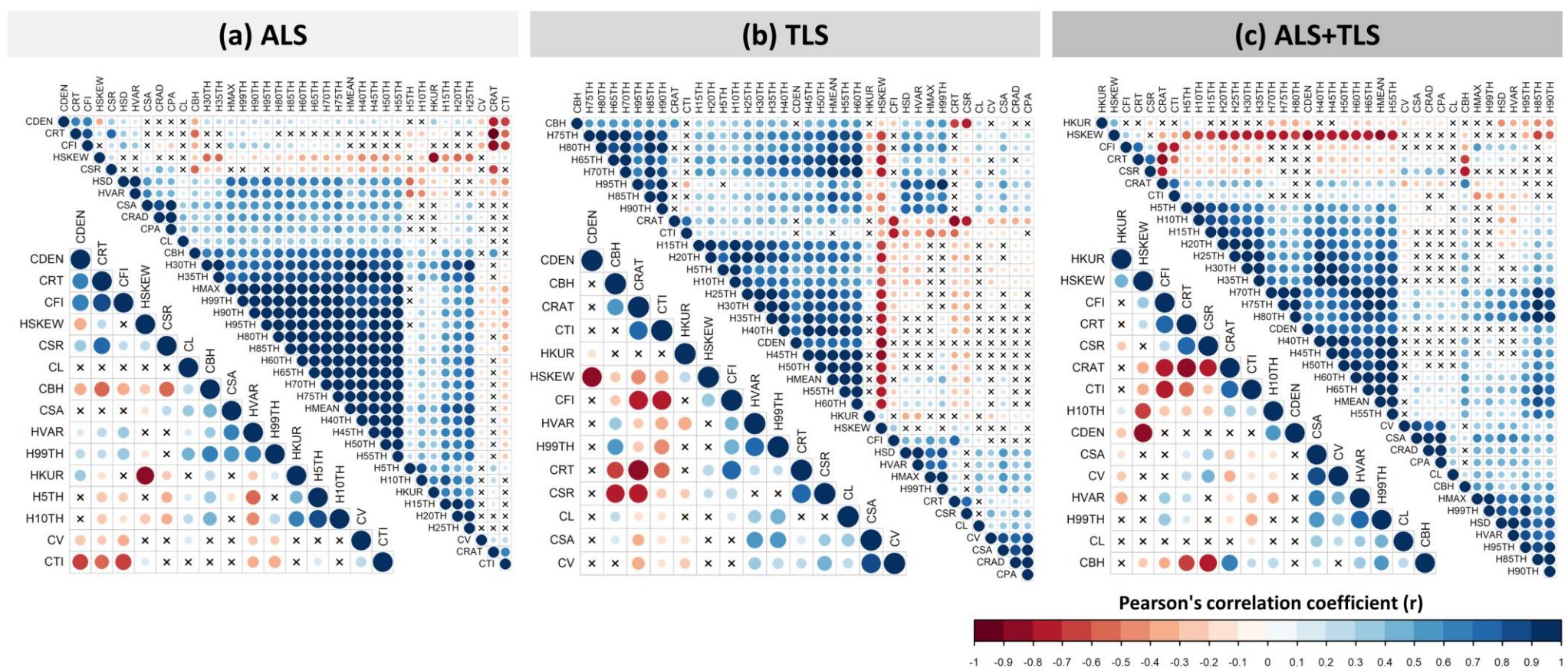

**Figure 5.** Pearson's correlation (r) among all the ALS (**a**), TLS (**b**), and ALS + TLS (**c**) derived crown-level metrics (upper triangle) and not highly correlated ($-0.9 < r < 0.9$) metrics (lower triangle). The "x" resents relationships where the r value is statistically equal to zero (*p*-value > 0.05).

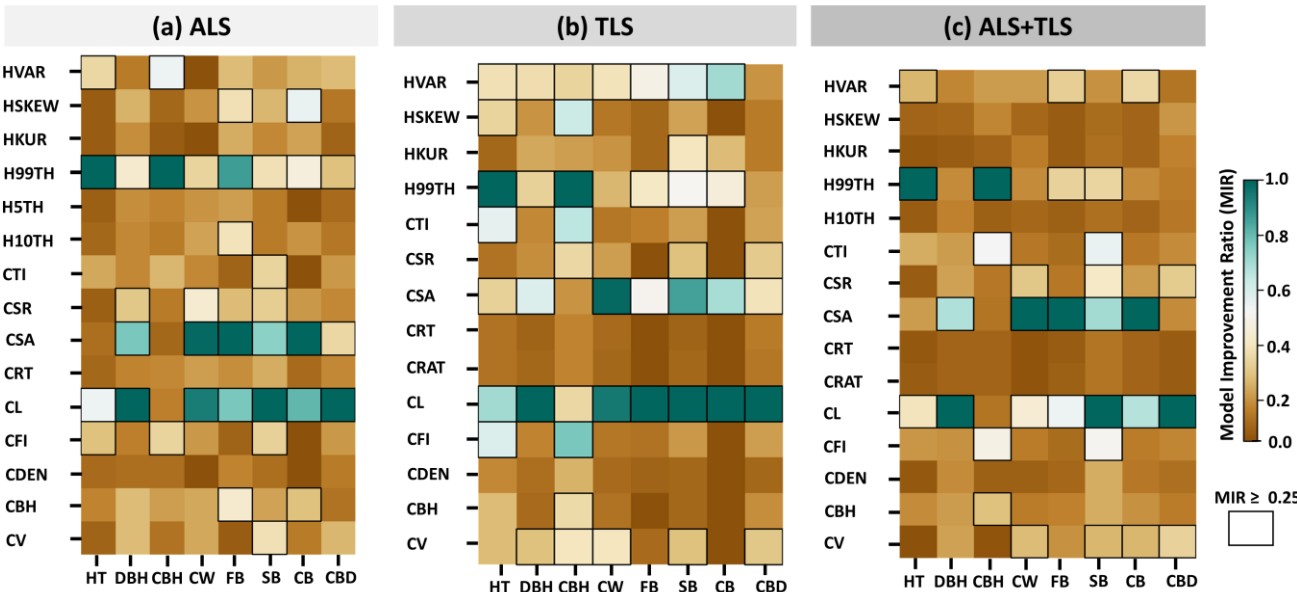

**Figure 6.** Mean model improvement ratio (MIR) of preselected metrics (i.e., metrics with Pearson's correlation between −0.9 and 0.9 with each other). Metrics with MIR ≥ 0.25 were selected for modeling.

For the ALS data, the number of selected metrics to train the RF model after the exclusion of highly correlated variables varied from three to seven (Figure 6). The model of CBH and SB included as predictors the lowest and largest number of metrics, respectively. For the TLS data, the number of selected metrics ranged from four (i.e., for CB and CBD) to nine (i.e., for CBH). For the fused dataset (ALS + TLS), the number of selected metrics ranged from two (i.e., for DBH) to seven (i.e., for SB). Regarding metric importance within the model, H99TH, CL, and CSA highlighted the importance in most models using the ALS, the TLS, and the ALS + TLS data.

*3.3. Random Forest Model Assessment*

RF model performance slightly varied across datasets and crown-level structure and fuel attributes. Overall, all models from the bootstrapping procedure performed well and produced $R^2$, RMSE, and bias estimates ranging from 0.80 to 0.98, from 1.06% to 9.31% and 0.75% to 7.63%, respectively (Table 5).

The final predictions derived from the mean of the 500 bootstrap runs were highly correlated to the observed values ($R^2$ between 0.80 and 0.98), with low to moderate errors (RMSE% between 4.56 and 50.99%) and bias close to zero (between 0.03% and −2.86%). By using the ALS data, the model of CBH depicted the highest $R^2$ (0.98), and the model of HT had the lowest RMSE (4.83%). For TLS and the fused dataset (ALS + TLS), the models for HT performed slightly better in comparison to the other modeled tree attributes. By using ALS data, predictions for SB had the smallest $R^2$ value (=0.80) and highest RMSE (51%), but the bias was low. The accuracy for this attribute was higher using the TLS ($R^2 \geq 0.88$ and RMSE < 39%) and the fused dataset (ALS + TLS). ALS performed better for the HT and CBH models. Accuracies in terms of bias were comparable among models and datasets, with values below absolute 3.00%. Overall, predictions for crown structural and fuel attributes tended to underestimate high values and overestimate low values (Figure 7).

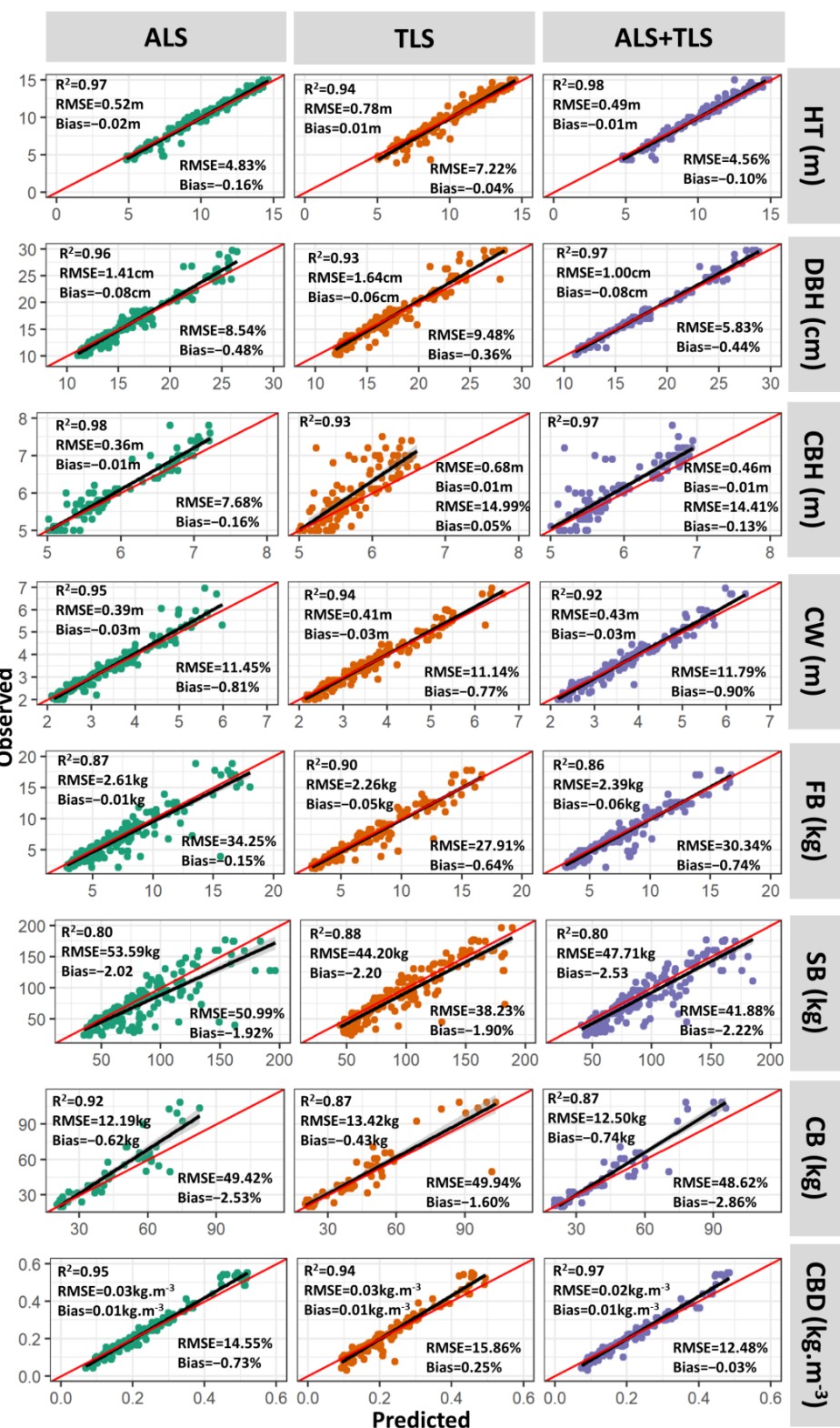

**Figure 7.** Scatter plots of observed versus predicted crown-level structure and fuel load attributes, RMSE, and bias across tree attributes and lidar datasets. The predicted values are derived from the average of the 500 bootstrap iterations. Diameter at breast height (DBH); tree height (HT); crown width (CW); canopy base height (CBH); foliage biomass (FB); steam branches biomass (SB); crown biomass (CB); crown bulk density (CBD). The diagonal lines represent the perfect match (1:1) between observed and predicted values.

**Table 5.** Bootstrap prediction accuracies (mean ± sd) from the random forest (RF) models (n = 500) in terms of $R^2$, absolute and relative root mean square error (RMSE), and bias. Tree height (HT); Diameter at breast height (DBH); crown width (CW); canopy base height (CBH); foliage biomass (FB); stem branches biomass (SB); crown biomass (CB); crown bulk density (CBD).

| Data | Forest Attributes | $R^2$ (Mean ± sd) | | RMSE (Mean ± sd) Absolute | | Relative (%) | | Bias (Mean ± sd) Absolute | | Relative (%) | |
|---|---|---|---|---|---|---|---|---|---|---|---|
| ALS | HT (m) | 0.97 | ±0.01 | 0.53 | ±0.11 | 4.94 | ±1.06 | −0.02 | ±0.09 | −0.15 | ±0.79 |
| | DBH (cm) | 0.94 | ±0.05 | 1.46 | ±0.60 | 8.78 | ±3.47 | −0.08 | ±0.24 | −0.48 | ±1.42 |
| | CBH (m) | 0.98 | ±0.01 | 0.37 | ±0.06 | 8.06 | ±1.42 | −0.01 | ±0.06 | −0.14 | ±1.18 |
| | CW (m) | 0.93 | ±0.05 | 0.39 | ±0.17 | 11.6 | ±4.62 | −0.03 | ±0.07 | −0.77 | ±1.89 |
| | FB (kg) | 0.86 | ±0.07 | 2.62 | ±0.82 | 34.19 | ±9.03 | 0.01 | ±0.43 | 0.28 | ±5.56 |
| | SB (kg) | 0.80 | ±0.09 | 51.30 | ±19.13 | 48.32 | ±14.72 | −1.74 | ±8.08 | −1.10 | ±7.49 |
| | CB (kg) | 0.90 | ±0.08 | 11.99 | ±5.80 | 47.42 | ±17.84 | −0.64 | ±2.03 | −1.86 | ±7.63 |
| | CBD (kg m$^{-3}$) | 0.95 | ±0.03 | 0.03 | ±0.01 | 14.15 | ±4.82 | 0.01 | ±0.01 | −0.73 | ±2.06 |
| TLS | HT (m) | 0.94 | ±0.02 | 0.79 | ±0.14 | 7.35 | ±1.35 | 0.01 | ±0.13 | 0.01 | ±1.19 |
| | DBH (cm) | 0.92 | ±0.04 | 1.67 | ±0.56 | 9.62 | ±2.99 | −0.07 | ±0.28 | −0.35 | ±1.59 |
| | CBH (m) | 0.92 | ±0.02 | 0.69 | ±0.09 | 15.39 | ±2.17 | 0.01 | ±0.11 | 0.16 | ±2.35 |
| | CW (m) | 0.93 | ±0.04 | 0.39 | ±0.15 | 10.81 | ±3.71 | −0.02 | ±0.06 | −0.62 | ±1.74 |
| | FB (kg) | 0.90 | ±0.05 | 2.21 | ±0.90 | 26.87 | ±9.31 | −0.05 | ±0.38 | −0.37 | ±4.50 |
| | SB (kg) | 0.87 | ±0.08 | 43.09 | ±17.03 | 36.82 | ±12.11 | −1.89 | ±6.97 | −1.26 | ±5.79 |
| | CB (kg) | 0.87 | ±0.08 | 12.83 | ±5.35 | 47.04 | ±15.66 | −0.40 | ±2.04 | −0.86 | ±7.28 |
| | CBD (kg m$^{-3}$) | 0.93 | ±0.03 | 0.03 | ±0.01 | 16.46 | ±3.29 | 0.01 | ±0.01 | 0.38 | ±2.58 |
| ALS + TLS | HT (m) | 0.97 | ±0.01 | 0.51 | ±0.11 | 4.70 | ±1.06 | −0.01 | ±0.08 | −0.07 | ±0.75 |
| | DBH (cm) | 0.97 | ±0.04 | 0.99 | ±0.57 | 5.69 | ±3.17 | −0.08 | ±0.17 | −0.45 | ±0.95 |
| | CBH (m) | 0.96 | ±0.01 | 0.49 | ±0.08 | 10.89 | ±1.80 | −0.01 | ±0.07 | −0.13 | ±1.59 |
| | CW (m) | 0.91 | ±0.05 | 0.43 | ±0.16 | 11.59 | ±4.02 | −0.03 | ±0.06 | −0.8 | ±1.67 |
| | FB (kg) | 0.87 | ±0.07 | 2.26 | ±0.95 | 28.60 | ±10.88 | −0.04 | ±0.36 | −0.38 | ±4.58 |
| | SB (kg) | 0.80 | ±0.11 | 46.65 | ±20.19 | 40.19 | ±15.23 | −2.71 | ±7.05 | −2.01 | ±5.94 |
| | CB (kg) | 0.88 | ±0.11 | 11.25 | ±6.72 | 42.60 | ±21.77 | −0.67 | ±1.92 | −2.04 | ±6.98 |
| | CBD (kg m$^{-3}$) | 0.96 | ±0.01 | 0.03 | ±0.01 | 12.99 | ±3.03 | 0.01 | 0.01 | −0.02 | ±2.02 |

## 4. Discussion

Maps of crown structure and fuel attributes are highly demanded by fire and forest managers in areas under frequently prescribed fire management for diverse applications such as simulations of fire behavior [33,65–70]. However, measuring such attributes in the field is difficult, costly, and prone to errors. To overcome this limitation, we developed an easily implementable and transferrable modeling framework using ALS and TLS data. Previous lidar-based studies showed the potential of lidar data for structure and fuel canopy modeling [71–76], but to our knowledge, this is the first study to assess the impact of ALS, TLS, and the fusion of both systems to crown-level structural and fuel metrics modeling in longleaf pine forest ecosystems.

Comparable tree detection errors were observed using both ALS and TLS systems (Table 4), which might be attributed to the point cloud densities of the lidar systems as well as to the open forest structure of the study site. The similarity in tree detection errors aligns with the results found by Huo et al. [77]. ITD algorithms are affected by the tree density: where stands are denser, and detection errors are higher [78,79]. Trees in longleaf and other managed southern pine savannas are sparse [80,81], in our case, around 347 trees per hectare, so the local maxima filter could distinguish a high proportion of tree tops regardless of the lidar dataset used. Utilizing only the ALS data at a relatively typical point density (7 points m$^{-2}$) was as effective as using the TLS data (68 points m$^{-2}$) or the ALS + TLS data (75 and points m$^{-2}$). Previous studies signaled the importance of lidar point density for tree detection but also pointed out that marginal improvements are achieved when more than five points m$^{-2}$ are used [82,83]. Additionally, the density plots of the lidar metric distribution showed no significant difference between the three datasets for the height metrics of the uppermost canopy (i.e., HMAX, H99TH) (Figure 4), indicating

that both scanners were able to sample tree tops with the same efficiency and estimate the tree attribute with comparable accuracy. This finding corresponds with other studies that found a strong correlation between tree heights measured from ALS and TLS [16,65,84].

ALS and TLS datasets produced significantly different tree height metrics (e.g., H5TH) distributions but similar crown-level metrics distributions (e.g., CL and CBH) (Figure 5), and TLS data depicted slightly better performance compared to ALS data in the prediction of crown structure and fuel attributes (e.g., FB, and SB) (Figure 7). That was expected based on the differences between the two systems. ALS returns come mostly from upper branches and the adaxial side of the leaves or needles, while returns from stems are relatively scarce [85,86]. In contrast, the number of returns coming from the stems and branches from the TLS system is relatively massive and can sample the geometry of needles and woody parts more easily and at different locations [87,88]. This difference in ALS and TLS operations is also suggested by Bazezew et al. [89]. These authors addressed in their study that ALS was more adequate for upper canopy detection, while TLS for stem measurements and lower canopy detection, TLS was more efficient.

TLS data have been traditionally collected from the ground using one or many scanning points per plot [90], but we used a TLS raised up to the canopy height to collect data from a single scanning position across three unusually large plots across a cumulatively large area. Such an approach is novel and was shown to enhance estimations of crown structure and fuel attributes. It should be noted, nevertheless, that a slightly higher accuracy was observed in the predictions of HT and CBH using the ALS data (Figure 7). This might be explained by this surveying strategy and the data processing. ALS can attain more points of the crown due to the data acquisition from a higher elevation on the top. Moreover, the terrain model derived from the ALS data was used to normalize both TLS and ALS data point clouds, so, likely, metrics of the uppermost height canopy (e.g., HMAX, H99, H90, etc.) that were used as predictors in these RF model were more accurately calculated when derived from ALS data.

The accuracy of the RF models using the fused dataset (ALS + TLS) in all modeled tree attributes was relatively good (e.g., $R^2 > 0.80$), and it was in agreement with other studies fusing both ALS and TLS data. Jung et al. [16], for instance, found correlations between predicted and observed values ($R^2$) of 0.94 for HT and 0.75 for CBH, while Giannetti et al. [91] found errors (RMSE) of 0.43 and 1.95 m for these attributes, respectively. Giannetti et al. [91] also found errors of 1.13 m for tree DBH, and Paris et al. [92] found $R^2$ of 0.86 for CW. However, it is important to point out that comparisons of accuracy among studies should be carefully considered due to differences in data collection, modeling approach, accuracy assessment, and evaluated forest type.

Despite this relatively good performance, we did not observe a gain in accuracy by fusing both lidar systems. The results of the fused dataset (ALS + TLS) were considerably similar to the results of both ALS and TLS data. Therefore, the use of either ALS or TLS would be sufficient to model the tree crown structural and fuel attributes in this longleaf pine forest ecosystem. Additionally, we believe that ALS data offers reasonable results regarding the difference in point density and data acquisition capabilities of the two lidar systems. This is operationally relevant because of the significantly larger area that can be surveyed with the airborne system, especially if compared to the TLS, which needs at least two scans captured from different perspectives or even more if stand density causes occlusion [93–95]. Additionally, other authors have developed studies with the fusion of different lidar datasets and have discussed how choosing the appropriate method will depend on the objective of the forester [96]. While there was an asynchrony in data collection that could potentially affect the accuracy of the models proposed here, given the slow growth rates of mature longleaf pine, and the metrics that we are evaluating in this study (crown-level structure and fuel metrics), we thoroughly expect that this impact is going to be minimal on our modeling.

We proposed a workflow to model tree/crown structure and fuel attributes, and we tested it in three specific inventory plots at Eglin AFB. Nevertheless, further research is

needed to assess the implications of our analysis across diverse study sites. Factors such as tree density, crown shape, tree overlaps, stem size, tree age, and species distribution affect stand and plot structures and, therefore, should be carefully considered while performing ITD and modeling individual tree attributes. Additionally, assessing the differences between using ALS and TLS data for modeling different forest types and stand conditions would inform decision making. For instance, it would provide insights on whether a specific data collection is required or if it is a justifiable tradeoff by looking into other external factors—such as the size of the study area, ease of access to the location, density of the forests, and cost of operations. Cost-effectiveness analyses of boom-lifted TLS data collections also remain to be assessed. Such analysis must consider the costs of measuring tree attributes and the costs of the TLS data collection, taking into account the covered area and the number of trees that can be accurately measured from each scanning point. It is possible that inexpensive UAV-lidar or mobile scanning lidar systems can collect similar data at lower cost and more effectively from multiple view perspectives compared to boom-lifted TLS.

## 5. Conclusions

In this study, we evaluated and compared the capability of ALS, TLS, and ALS + TLS fusion in assisting with crown structure and fuel attribute estimations in a longleaf pine forest ecosystem and demonstrated that lidar could enhance structure and fuel load modeling at the tree level. The workflow involved individual tree detection from lidar point clouds and the development of predictive models of crown-level structure and fuel and tree attributes. Overall, both ALS- and TLS-based models showed relatively good performance in the prediction of the tree attributes. Given that TLS measurements are less practical and more expensive, our comparison suggests that ALS measurements are still reasonable for many applications, and their usefulness is justified. Our analyses demonstrate the applicability of lidar for modeling crown structure and fuel attributes and provide insights for advancing forest fuel and fire management strategies. Further research is needed to assess the operational and economic implications of these results for forest monitoring, fuel management, and fire activity planning at a broader scope.

**Author Contributions:** Conceptualization, K.D.R., C.A.S. and D.N.C.; methodology, K.D.R., C.A.S., D.N.C., C.K. and R.V.L.; software C.A.S., D.N.C., C.K. and R.V.L.; validation, C.A.S., D.N.C., C.K. and R.V.L.; formal analysis, K.D.R., C.A.S. and D.N.C.; investigation, K.D.R., C.A.S., D.N.C., C.K. and R.V.L.; resources, C.A.S.; data curation, C.A.S., D.N.C., C.K., R.V.L. and A.T.H.; writing—original draft preparation, K.D.R., C.A.S., D.N.C., C.K., R.V.L., R.P., N.S.-L., D.R.A.d.A., A.C., E.R., S.J.P., A.T.H., M.M., J.X. and M.B.S.; writing—review and editing, K.D.R., C.A.S., D.N.C., C.K., R.V.L., R.P., N.S.-L., D.R.A.d.A., A.C., E.R., S.J.P., A.T.H., M.M., J.X., J.W.A. and M.B.S.; visualization, C.A.S.; supervision, C.A.S.; project administration, C.A.S.; funding acquisition, C.A.S., S.J.P. and A.T.H. All authors have read and agreed to the published version of the manuscript.

**Funding:** This research was funded by the Department of Defense's Strategic Environmental Research and Development Program (SERDP), grant numbers: RC19-1064, RC20-1346 and RC20-1025, by the 2012 RxCADRE Project (11-2-1-11), and the EMS4D (22-2-02-15) of the Joint Fire Science Program.

**Data Availability Statement:** Tree data are available for download from [42]; ALS data are available for download from [45]; TLS data are available for download from [46].

**Acknowledgments:** This research was supported by the U.S. Department of Agriculture, Forest Service, Rocky Mountain Research Station. The findings and conclusions in this publication are those of the au-thors and should not be construed to represent any official USDA or U.S. Government determi-nation or policy. D.R.A.d.A. was supported by the Sao Paulo Research Foundation (#2018/21338-3). We would like to express our sincere thanks to the funding agency.

**Conflicts of Interest:** The authors declare no conflict of interest.

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
