# Peer review of "Crown-Level Structure and Fuel Load Characterization from Airborne and Terrestrial Laser Scanning in a Longleaf Pine (Pinus palustris Mill.) Forest Ecosystem"

_remotesensing, doi:10.3390/rs15041002_

Round 1

Reviewer 1 Report

The submitted manuscript entitled “Crown-level structure and fuel load characterization from airborne and terrestrial laser scanning in a longleaf pine (Pinus 3 palustris Mill.) forest ecosystem by Diego da Rocha et al. tests the applicability of ALS, TLS, and the fusion of the two when quantifying forest structure metrics. Though the findings show TLS outperforming ALS, the fact that ALS provides suitable predictions of crown-level attributes will allow for the assessment these attributes at larger extents in longleaf pine systems. The manuscript is well written with informative figures and tables that make this study easy to comprehend.

Overall, I have very few comments or critiques for the manuscript. There are a few miniscule comments that are outlined below with line numbers. The only larger comment I have is regarding the plot selection. Because there are only three plots in this study, there should be more detail on how they were selected. I am left wondering if these three plots encompass the variability seen in longleaf pine systems, and how that may impact the ALS accuracies. Essentially, there should be some more discussion or clarification on how applicable these models are to longleaf pine systems at larger extents.  

Line 161: Figure 1 needs to be a higher resolution. Additionally, the CHM is difficult to see. Maybe the author could change the colors or breaks within the gradient.

Line 166: How were plots selected?

Line 227: What is the AD in CRAD? This could be an overlook on my end.

Line 243: “…inventoried in the field for each plot (Table 1)” I believe this should reference Table 4

Line 245 (Table 3): Why is meters represented as M instead of m for crown base height and crown length?

Line 341 (Figure 4): Could go in supplemental.

Lines 395-397 (Table 6 & Figure 6): If Table 6 and Figure 6 represent the same values, put one in supplemental.

Author Response

Authors: Thank you for your comments and your reflections on our paper. We believe your contributions have offered great value to this manuscript and we are ecstatic with the opportunity to address the questions you have raised here. We are certain that now we have an even stronger manuscript, and we are extremely thankful for your important remarks. Our detailed responses will be uploaded as a pdf attachment.

Reviewer 2 Report

Abstract: Line 31, I think UAS-LiDAR also gaining much attention in the current market; therefore, the word currently should be replaced with the word "frequently".

1. Introduction

Though the introduction provides significant information on the subject being addressed. My recommendation is to break down the introduction part into sub-sections as follows.

1. Introduction

a general overview of the proposed study along with a literature review

1.1. Related work section specifically highlighting the previous studies of similar nature.

1.2. Contributions made in this study, particularly focusing on the unique findings. The authors are mentioning two things. Forest fire modeling is one thing and extracting tree-level or stand-level parameters is another. Please only mention the contribution made in this paper in section 1.2. and everything else e.g., applications should go to the introduction or related work part. At this stage, too many things lead to confusion such as fire burn, fuel loading, tree parameters, machine learning, statistical modeling, etc. The mentioned metrics: DBH, HT, CBH, CW, FB, SB, CB, and CBD, are they forest structural attributes, or fire models attributes.  Please justify your real objective. 

2. Materials (move methods to next section 3. Methods)

2.1 Study area

Figure 1.  As authors possess TLS and ALS datasets, it would be much better to show some LiDAR 3D transects (2D vertical profiles): One in North-South and one in East-West directions. One transects with ALS and one with TLS. Then the text mentioned in lines 157-160 can be justified with reference to modified Figure 1. Authors can occupy the blank space left between lines 160-161 with some LiDAR datasets as a description of the study area. Please check this paper as a guiding principle.

Comparing tree attributes derived from quantitative structure models based on drone and mobile laser scanning point clouds across varying canopy cover conditions - ScienceDirect

Figure 1. For captions, please use the style (a), (b), and (c). I could not find D.

This should follow by

2.1. ALS survey

Line 201 MPiA ? what is it, please provide ref.  How much area is covered by ALS flights. There is no detail description of acquire data and its horizontal and vertical accuracies. This section should cover all the aspect of collected LiDAR data inclusive, PRR, flight altitude, beam divergence, scanning mode etc. Table 2. Please use the scientific notation of kHz instead of Hz.

2.2. TLS survey

Compared with ALS, TLS usually performed with fixed stations otherwise its MLS. How many stations was installed in the survey area? How much area was covered by each scan-station? The scan station’s locations can be marked on Figure 1.

2.2. Field Data collection.

Please provide the ref. Line 168 and 170.

A better approach is to add a table with the following attributes

Site ID, Estimated parameter, sensor/instrument, method. Table 1 should be revised.

3. Methods

Please first add a detail workflow how the acquired datasets were processed following the above-mentioned paper. Then each step should follow the same order as mentioned in the workflow diagram.

Line 204-209 should be relocated to method sections. Please provide the details of certain things how it was accomplished e.g., ground points were classified using progressive triangulated irregular network densification algorithm. Authors do not provide information about how TLS scan stations were registered to create a complete dataset. Also, authors lack the information on how the ALS and TLS were registered to create a fused ALS+TLS datasets given the fact that both systems capture data in different coordinates systems.

Sections 2.3., 2.4., should come under the methods sections following a schematic flow diagram as mentioned earlier.

Section 3.3. evaluation metrics:

Detail all the equations used for evaluating the results in this section. All equations should be properly cited.

3. Results

Line 296, the word tallied somewhat confusing, find an alternative way to describe the comparison.

Line 300-301, Table 4 and Figure 2. why TLS gives higher error range as most of the studies suggest that the TLS is richer in information than TLS given the fact that TLS can capture more structural attributes than ALS.

What function does Figure 3c1-c3 serve and how it was generated? Figure a1, a2, a3, b1, b2, b3, c1,c2,c3 is not mentioned in the text. Ref. section 3.1. Figure 3. Is not well explained in the next section. The test mentioned “KS” first needs to be mentioned in the method section or sub-section evaluation metrics.

Please arrange the results in a similar fashion as mentioned in the data and method sections. Figure 4. Is not readable in the present format, maybe occupy more space by rearranging the layout. My recommendation is to use a Portrait layout instead of Landscape by arranging the Figure as follows ALS, TLS, and ALS+TLS.

Overall, Data and method sections require extensive editing as per provided guidelines. The results section can further be improved remember to consult the above-mentioned paper.

References.

Ref. 42 should be the given below reference for progressive TIN densification filtering. Please make sure citations are properly used in the text as I cannot check every citation and not giving the right credit to original authors can therefore be misleading.

X. Zhao, Q. Guo, Y. Su, and B. Xue, “Improved progressive TIN densification filtering algorithm for airborne LiDAR data in forested areas,” ISPRS J. Photogramm. Remote Sens., vol. 117, pp. 79–91, Jul. 2016.ences

Best of Luck

Author Response

Authors: We are deeply thankful for all your contributions to our manuscript. We recognize your effort in making this manuscript more robust and we are glad you took your time to give such significant contributions. Your feedback is deeply appreciated, and it gives us many meaningful insights that have, without a doubt, helped us in enhancing our work and presenting our information in a more concise and detailed way. We can assure you that we have carefully addressed all of your concerns to the best of our ability, and we have made major changes based on your comments. We recognize that your contributions were of crucial importance, and we are overjoyed that you have helped us in this process. Thank you. Our detailed responses will be uploaded as a pdf attachment.

Reviewer 3 Report

Dear colleagues!

The authors' research has a clear applied character and can be used to assess the successive processes of forest ecosystems, which is very important in the face of rapid climate change on the planet.

The authors proposed 2 laser scanning methods and their combinations of both for describing crown structure and fuel attributes of Pinus palustris Mill. at Eglin AFB.

The authors supplemented the instrument measurements with a real assessment of the taxation indicators of trees.

There is competent and persuasive introduction. In chapter 2, the authors described in detail the research methods and gave detailed model diagram and calculation formulas.

The using of modern statistical methods inspires confidence in the results of the research.

The authors also described in detail the results of the work, illustrated their judgments, and conducted a discussion of the results.

The conclusions of the research are beyond doubt.

For a small improvement of the manuscript I ask the authors to make the following additions:

1. In the methodology I ask you to specify the number of trees that have been subjected to the taxation analysis.

2. You conducted field observations in 2017, but in your work you provide remote sensing data for 2012. I can assume an objective reason for this asynchrony. I ask you to indicate how much this gap of 5 years played a role in possible negative changes.

3. I ask you to deepen the discussion of the results of the study. The list of literature sources looks insignificant.

Author Response

Authors’ Response: We deeply appreciate your comments about the quality of the paper, as well as the specific improvement suggestions. We have carefully reviewed all of your suggestions and we have meticulously addressed them all. We are certain that we now have a more well-rounded and robust article and we would like to thank you for these contributions as you have helped us improve our manuscript. Our detailed responses will be uploaded as a pdf attachment.

Reviewer 4 Report

General comments

This manuscript describes a verification that lidar-based systems can predict the canopy structure of forests.  Although this is unlikely to be the first study to do this, I believe this analysis is important to publish.  I found the overall description of the study clear, that the methods were suitable, and the conclusions were justified.

My main concern is two-fold.  First, the manuscript is very imprecise in some of the terms used.  For example, growth could mean vegetative growth in a general sense, but technically growth is the increase of something over time.  So referring to structure as growth is confusing to the more literally minded. Similarly, recording is not the same as locating. Second, the manuscript is quite wordy and I found a number of places where three to four words could be reduced to one.  Examples are listed below, but I encourage the authors to remove excess words and to be more precise in the wording they do use.

Specific comments (line)

35 It is not clear what defined means here.  It is not clear how three plots could define a landscape. They might be used to characterize it or describe it. 

26 Was there one plot per condition? Or three plots per condition?

44 all three lidar related data sets. Not clear which data is being referred to as there is lidar and non-linar related data.

51 which three? The lidar-related ones? There were more than three methologies in the analysis.

55 The problem the wording here is that it vague. Reasonable is in the eye of the beholder. Same thing with usefulness.

65 While I agree that growth is important, the structure is also important to how a forest functions.  Given that the analysis is really about structure, would it not make sense to include structure here?

68 “that are happening” could be replaced by “occurring”.  Also why are they our forests? Is that needed to make the point?

76 what does “help keep a check on” mean exactly?  Reduce? Or ameliorate?

77 could one not write “many” or “numerous” instead of “a wide range of”.  I would encourage the authors to replace many of these wordy, vague phrases with single words wherever possible.

106 Are these the issues or are the issues related to these factors?  There is a difference in terms of logic.

110 Does this mean that maximum height is reached when trees are 25 cm DBH? And height growth can’t be 25 m.  The maximum height might be 25 m, but the growth rate has to be great deal smaller and also involve the dimension of time.

112 This does not entirely make sense. The limitations are for ALS not TLS.  As written that is not clear.

135 Is the “herein” needed?

139 It is not clear what stretch means.

150 not clear why DBH is a crown structure variable.  It is a structural variable, but not actually a crown variable as implied here.

157 the meaning of sparse is not clear.  Does this mean occasional? Or are the trees themselves sparse in terms of foliage?

166 “within” would make more sense than “at” because a plot is two dimensional.  If they were points then at would make more sense.

167 “located” would be better than “recorded”.  This is an example of the imprecision of language throughout this manuscript.

Author Response

Authors’ Response: Thank you for the comments. We are glad to hear that you agree with the importance of the analysis addressed in this manuscript. We have carefully reviewed the concerns and questions you have raised in your review, and we can confidently say that we have made major changes based on your suggestions that have improved the quality of this manuscript. We appreciate the time and effort you have put in reviewing this study and we are thankful for the insightful feedback. Our detailed responses will be uploaded as a pdf attachment.

Round 2

Reviewer 2 Report

Authors have addressed most of the comments and added important information. I recommend accepting in present form.